# Tyrosinase Inhibitors Derived from Chemical Constituents of *Dianella ensifolia*

**DOI:** 10.3390/plants11162142

**Published:** 2022-08-18

**Authors:** Yu-Chang Chen, Sheng-Han Su, Jheng-Cian Huang, Che-Yi Chao, Ping-Jyun Sung, Yih-Fung Chen, Horng-Huey Ko, Yueh-Hsiung Kuo

**Affiliations:** 1School of Pharmacy, College of Pharmacy, Kaohsiung Medical University, Kaohsiung 807, Taiwan; 2Department of Chemistry, National Taiwan University, Taipei 106, Taiwan; 3Department of Food Nutrition and Health Biotechnology, Asia University, Taichung 413, Taiwan; 4National Museum of Marine Biology and Aquarium, Pingtung 944, Taiwan; 5Graduate Institute of Natural Products, College of Pharmacy, Kaohsiung Medical University, Kaohsiung 807, Taiwan; 6Department of Medical Research, Kaohsiung Medical University Hospital, Kaohsiung Medical University, Kaohsiung 807, Taiwan; 7Department of Fragrance and Cosmetic Science, College of Pharmacy, Kaohsiung Medical University, Kaohsiung 807, Taiwan; 8Drug Development and Value Creation Center, Kaohsiung Medical University, Kaohsiung 807, Taiwan; 9Department of Chinese Pharmaceutical Sciences and Chinese Medicine Resources, College of Pharmacy, China Medical University, Taichung 404, Taiwan; 10Chinese Medicine Research Center, China Medical University, Taichung 404, Taiwan; 11Department of Biotechnology, Asia University, Taichung 413, Taiwan

**Keywords:** *Dianella ensifolia*, Asphodelaceae, flavan, tyrosinase

## Abstract

*Dianella* *ensifolia* is a perennial herb with thickened rhizome and is widely distributed in tropical and subtropical regions of Asia, Australia, and the Pacific islands. This plant has the potential to be used as a source of herbal medicine. This study investigated further phytochemistry and tyrosinase inhibitory effect of some constituents isolated from *D. ensifolia*. Four new flavans, (2*S*)-4’-hydroxy-6,7-dimethoxyflavan (**1**), (2*S*)-3’,4’-dihydroxy-7-methoxy-8-methylflavan (**2**), (2*S*)-2’-hydroxy-7-methoxyflavan (**3**), and (2*S*,1′*S*)-4-hydroxy-4-(7-methoxy-8-methylchroman-2-yl)-cyclohex-2-enone (**4**), together with 67 known compounds, including 10 flavans (**5**–**14**), 5 flavanones (**15**–**19**), 3 flavone (**20**–**22**), 5 chalcones (**23**–**27**), 3 chromones (**28**–**30**), 15 aromatics (**31**–**45**), 7 phenylpropanoids (**46**–**52**), one lignan (**53**), 7 steroids (**54**–**60**), one monoterpene (**61**), one diterpene (**62**), 4 triterpenes (**63**–**66**), a carotenoid (**67**), 2 alkaloids (**68** and **69**), and 2 fatty acids (**70** and **71**) were isolated from *D. ensifolia*. Their structures were elucidated on the basis of physical and spectroscopic data analyses. Moreover, compounds **1**–**4**, **8, 10**–**15, 20, 21,** and **41** were evaluated for their mushroom tyrosinase inhibitory effect. Compounds **11** and **14** strongly inhibited mushroom tyrosinase activity with IC_50_ values of 8.6 and 14.5 μM, respectively.

## 1. Introduction

The *Dianella* Lam. ex Juss. genus includes perennial herbs (or subshrubs), glabrous, and rhizomatous with aerial stems. It comprises about 40 species with accepted names, mainly distributed in tropical Asia, Australia, and the southern Pacific islands [1]. The *Dianella* genus previously belongs to Liliaceae but has been revised to Asphodelaceae, recently [1,2]. Only one species, *Dianella ensifolia* (L.) DC, is native to Taiwan [3]. *D. ensifolia* is used as herbal medicine, such as topical medicines for ferunculosis, abscesses, lympharingitis, tuberculosis lymphadenitis, tinea, traumatic injuries, and wounds. The plant can also be taken internally for the treatment of dysentery, dysuria, leucorrhoea, blanorrhoea, and fatigue [4]. In Taiwan, the indigenous people use its roots for the treatment of abdominal pain; roots and leaves for treating poisonous snakebite [5]. Previous studies of *D. ensifolia* identified various phytochemical constituents, such as quinones [6,7], phenolics, steroids [8], flavonoids [8,9,10], glycosides [11], triterpenoids [12,13], and diarylpropanes [14,15,16]. Some of these constituents exhibited biological activities, including antioxidative [12,16], antibacterial [12], antivirus [12,17,18], anthelmintic [10], antitumor/anticancer [10,13], and anti-tyrosinase/melanogenesis activities [15,16,19]. Given the promise of a literature survey, an investigation was carried out to search for additional valuable bioactive constituents from the whole plant of *D. ensifolia*.

Melanin is one of the most widely distributed pigments and is found in fungi, bacteria, plants, and animals. It is the main protective pigment found in the hair, skin, and eyes of a human. The role of melanin is to protect the skin against ultraviolet (UV) damage by absorbing UV sunlight and removing reactive oxygen species (ROS) [20]. Melanin is formed through a series of oxidative reactions involving the amino acid tyrosine in the presence of tyrosinase. Tyrosinase (EC 1.14.18.1) is a membrane-bound, multifunctional copper-containing polyphenol oxidase (PPO) enzyme, involved in the initial steps of melanin biosynthesis in living organisms [21,22]. The enzyme catalyzes the hydroxylation of l-tyrosine to l-3,4-dihydroxyphenylalanine (l-DOPA) and l-DOPA to dopaquinone, which is the rate-limiting step of melanin synthesis and can cause unusual melanin pigments accumulation in the outermost layer of the skin [23,24]. Excessive production of melanin resulted in different dermatological disorders such as aging spots, wrinkles, melasma, freckles, lentigo, ephelides, post-inflammatory melanoderma, nevus, and melanoma [25]. In addition, a recent investigation suggests that abnormal melanogenesis disorders are related to some neurodegenerative diseases, such as Alzheimer’s, Parkinson’s, and Huntington’s diseases [26]. Therefore, there is a great need for melanogenesis inhibitors to develop treatment or prevention of hyperpigmentation disorders. Tyrosinase inhibitors typically work by chelating with copper within the active site of tyrosinase, obviating the substrate-enzyme interaction, or disrupting the ensuing electrochemical oxidation process [23]. Since it has been observed that flavonoids, stilbenes, and phenolics manifest tyrosinase inhibitory activities, it encourages us the continuing search for more bioactive constituents from Formosan medicinal plants. Accordingly, this investigation of *D. ensifolia* products focused on identifying new tyrosinase inhibitors that could be whitening candidates. Here in this article, the structure elucidation of four new flavans and the results of the tyrosinase inhibitory effect are reported.

## 2. Results and Discussion

### 2.1. Isolation and Structure Elucidation of Compounds **1**–**4**

The methanolic extract of roots of *D. ensifolia* was partitioned into ethyl acetate (EtOAc)- and water-soluble layers. Four new flavans (**1**–**4**) (Figure 1) and 41 known compounds including 8 flavans (**5**–**12**), 2 flavanones (**15**–**16**), a flavone (**20**), 4 chalcones (**23**–**26**), 2 chromones (**28**–**29**), 11 aromatics (**31**–**41**), 4 phenylpropanoids (**46**–**49**), one lignin (**53**), 7 steroids (**54**–**60**), and a monoterpene (**61**) (Appendix A) were isolated from the EtOAc-soluble layer of methanolic extract of roots.

The methanolic extract of the aerial part of *D. ensifolia* was partitioned into EtOAc- and water-soluble layers. The new flavan (**1**) and 48 known compounds including 7 flavans (**8**–**14**), 5 flavanones (**15**–**19**), 2 flavones (**21**–**22**), 2 chalcones (**26**–**27**), 2 chromones (**29**–**30**), 10 aromatics (**36**–**45**), 4 phenylpropanoids (**49**–**52**), lignin (**53**), 4 steroids (**54**–**57**), a monoterpene (**61**), one diterpene (**62**), 4 triterpenes (**63**–**66**), a carotenoid (**67**), two alkaloids (**68**–**69**), and two fatty acids (**70**–**71**) (Appendix A) were isolated from the EtOAc-soluble layer of methanolic extract of the aerial part.

Compound **1** was isolated as yellowish oil. Its molecular formula, C_17_H_18_O_4_, was determined by HREIMS ([M]^+^, 286.1209). The ^1^H- and ^13^C-NMR spectral data (Table 1 and Table 2) of **1** were similar to that of 6,4’-dihydroxy-7-methoxyflavan [27], except that one hydroxy group at C-6 in 6,4’-dihydroxy-7-methoxyflavan was replaced by a methoxy group in **1**. The ^1^H-NMR spectrum (Table 1 and Appendix A) of **1** showed five aliphatic protons at δ 2.05, 2.14, 2.70, 2.92, and 4.92, typical of pyran nucleus protons of a flavan, one set of A_2_B_2_ system phenyl protons [δ 6.83 (2H, d, *J* = 8.4 Hz, H-3’ and H-5’), 7.28 (2H, d, *J* = 8.4 Hz, H-2’ and H-6’)], two singlet aromatic protons [δ 6.46 (1H, s, H-8), 6.57 (1H, s, H-5)], two methoxy singlet at δ 3.80 and 3.82, and one hydroxy group at δ 5.05. The NOESY spectrum of **1** (Figure 2 and Appendix A) showed protons at δ 3.80 and 3.82 owing correlations with 6.46 (H-8) and 6.57 (H-5) respectively suggesting a methoxy group (δ 3.80) at C-7 and a methoxy group (δ 3.82) at C-6. Because **1** showed negative optical rotation {${\left[\alpha \right]}_{D}^{25}$ −17.1° (*c* 0.01, MeOH)}, C-2 possesses as *S*-configuration [28]. According to the above evidence, the structure of **1** was elucidated as (2*S*)-4’-hydroxy-6,7-dimethoxyflavan, which was further confirmed by COSY (Appendix A), NOESY (Figure 2 and Appendix A), ^13^C-NMR (Appendix A), DEPT, HMQC (Appendix A), and HMBC (Figure 2 and Appendix A) experiments.

Compound **2** was isolated as a brown oil. Its molecular formula, C_17_H_18_O_4_, was determined by HREIMS ([M]^+^, 286.1207). The ^1^H-NMR spectrum (Table 1 and Appendix A) of **2** showed five aliphatic protons at δ 1.90, 2.16, 2.72, 2.92, and 4.96, typical of pyran nucleus protons of a flavan, one 3,4-dihydroxyphenyl group [δ 6.84 (2H, s, H-5’ and H-6’), 6.95 (1H, s, H-2’), 5.23 (br s, OH), 5.25 (br s, OH)], two *ortho* coupled doublets (*J* = 8.0 Hz) at δ 6.44 and 6.85, a methoxy singlet at δ 3.80, and one methyl singlet at δ 2.20. The NOESY experiment of **2** (Figure 3 and Appendix A) showed that protons at δ 6.85 correlated with δ 6.44 and 2.72 (H-4), δ 6.85 and 6.44 were assigned to H-5 and H-6, respectively. The methoxy singlet (δ 3.80) owning correlation with H-6 suggested the methoxy group at C-7. Compound **2** showed negative optical rotation {${\left[\alpha \right]}_{D}^{25}$ −15.8° (*c* 0.01, MeOH)}, which means C-2 has the *S*-configuration [28]. According to the above evidence, the structure of **2** was elucidated as (2*S*)-3’,4’-dihydroxy-7-methoxy-8-methylflavan, which was further confirmed by COSY (Appendix A), NOESY (Figure 3 and Appendix A), ^13^C-NMR (Appendix A), DEPT, HMQC (Appendix A,) and HMBC (Figure 3 and Appendix A) experiments.

Compound **3** was isolated as a brown oil. Its molecular formula, C_16_H_16_O_3_, was determined by HREIMS ([M]^+^, 256.1104). The ^1^H-NMR spectrum (Table 1 and Appendix A) of **3** showed five aliphatic protons at δ 2.26, 2.80, 2.96, and 5.18, typical of pyran nucleus protons of a flavan, one set of ABX system phenyl protons [δ 6.46 (d, *J* = 2.4 Hz), 6.52 (dd, *J* = 8.4, 2.4 Hz), 7.00 (d, *J* = 8.4 Hz)], four aromatic protons at δ 6.90, 6.91, 7.14, and 7.20, a methoxy singlet at δ 3.75. Since the NOESY experiment of **3** (Figure 4) showed that protons at δ 7.00 correlated with δ 6.52 and H-4, δ 7.00 and 6.52 were assigned to H-5 and H-6, respectively. The methoxy singlet owning correlation with H-6 and δ 6.46 suggested the methoxy group at C-7 and δ 6.46 was assigned to H-8. ^1^^3^C-NMR spectrum (Table 2 and Appendix A) of **3** showed three oxygenated quaternary C-atoms (δ 154.4, 154.7, 159.1) suggesting a hydroxyl group on B-ring. The NOESY correlations (Figure 4 and Appendix A) of four aromatic protons (δ 6.90, 6.91, 7.14, and 7.20) suggested the hydroxy group at C-2′. An *S*-configuration at C-2 of **3** was due to the negative optical rotation {${\left[\alpha \right]}_{D}^{25}$ −26.0° (*c* 0.01, MeOH)} [28]. According to the above evidence, the structure of **3** was elucidated as (2*S*)-2’-hydroxy-7-methoxyflavan, which was further confirmed by COSY (Appendix A), NOESY (Figure 4 and Appendix A), ^13^C-NMR (Appendix A), DEPT, HMQC (Appendix A), and HMBC (Figure 4 and Appendix A) experiments.

Compound **4** was isolated as a colorless oil. Its molecular formula, C_17_H_20_O_4_, was determined by HREIMS ([M]^+^, 288.1363). The ^1^H-NMR spectrum (Table 1 and Appendix A) of **4** was similar to that of **2**, except the typical aromatic proton signals of the B-ring were absent. Instead, four aliphatic protons at δ 2.16, 2.49, 2.81, and two vinylic hydrogens [6.06 (d, *J* = 10.4 Hz, H-3’), 7.09 (dd, *J* = 10.4, 1.2 Hz, H-2’)] provided evidence of a cyclohexenone non-aromatic B-ring. Moreover, an additional carbonyl group at δ 198.2 (C-4’) and conjugated vinylic carbons at δ 129.5 (C-3’) and 148.7 (C-2’) in the ^1^^3^C-NMR spectrum (Table 2 and Appendix A) of **4** provided strong support of this cyclohexenone moiety. HMBC correlations (Figure 5 and Appendix A) between H-2’ and the carbon at δ 80.5 (C-2), 198.2 (C-4’), 70.5 (C-1’), and 30.9 (C-6’) and between H-3’ and carbons at δ 70.5 (C-1’) and 33.9 (C-5’) supposed the connectivity between the non-aromatic B-ring and C-ring. According to the above evidence, the structure of **4** was elucidated as 4-hydroxy-4-(7-methoxy-8-methylchroman-2-yl)-cyclohex-2-enone, which was further confirmed by COSY (Appendix A), NOESY (Figure 5 and Appendix A), ^13^C-NMR (Appendix A), DEPT, HMQC (Appendix A), and HMBC (Figure 5 and Appendix A) experiments. The CD spectrum (Figure 6: orange curve) of **4** showed two positive Cotton effects around 392 and 334 nm, similar to the calculated ECD spectrum for (2*S*,1′*S*)-**4** (Figure 6: purple curve). Therefore, the absolute configuration of **4** was decided as (2*S*,1′*S*)-4-hydroxy-4-(7-methoxy-8-methylchroman-2-yl)-cyclohex-2-enone.

The known compounds including 10 flavans, (2*R*,3*R*)-3,4’-dihydroxy-7-methoxy-8-methylflavan (tupichinol A) (**5**) [29], (2*S*)-7,3’-dihydroxy-4’-methoxyflavan (**6**) [30], (2*S*)-4’-hydroxy-5,7-dimethoxy-8-methylflavan (**7**) [31], (2*S*)-7,4’-dihydroxyflavan (**8**) [28], (2*S*)-4’-hydroxy-7-methoxyflavan (**9**) [31], (2*S*)-7,4’-dihydroxy-8-methylflavan (**10**) [32], (2*S*)-2’,4’-dihydroxy-7-methoxy-8-methylflavan (**11**) [9], (2*S*)-3’,5’-dihydroxy-7,4’-dimethoxy-8-methylflavan (**12**) [33], (2*S*)-7,3′-dihydroxy-6,4’-dimethoxyflavan (**13**) [34], (2*S*)-2’,4’-dihydroxy-7-methoxyflavan (**14**) [8]; 5 flavanones, (2*S*)-liquiritigenin (**15**) [35], (2*S*)-farrerol (**16**) [36], (2*S*)-liquiritigenin 4’-methyl ether (**17**) [28], (2*S*)-5,7-dihydroxy-4’-methoxy-6,8-dimethylflavanone (**18**) [37], cyrtominetin (**19**) [38]; 3 flavones, luteolin (**20**) [39], apigenin (**21**) [39], syzalterin (**22**) [36]; 5 chalcones, broussonin A (**23**) [40], 1-(2,4-dimethoxyphenyl)-3-(4-hydroxyphenyl)propane (**24**) [41], 4,2’-dihydroxy-4’-methoxychalcone (**25**) [42], 4,2’,4’-trihydroxychalcone (**26**) [42], kukulkanin B (**27**) [42]; 3 chromones, noreugenin (**28**) [43] and 5,7-dihydroxy-2,8-dimethylchromen-4-one (**29**) [44], 5,7-dihydroxy-2-tricosylchromone (**30**) [45]; 15 aromatics, 1*H*,3*H*-5,7-dimethoxyisobenzofuran-l-one (**31**) [46], syringic acid (**32**) [47], 2,4-dimethoxy-6-methylbenzoic acid (**33**) [48], methyl 2,6-dihydroxy-3,4-dimethylbenzoate (**34**) [49], methyl 3,4-dihydroxybenzoate (**35**) [50], 3,5-dimethoxy-4-hydroxybenzaldehyde (**36**) [51], vanillic acid (**37**) [52], methyl 2,4-dimethoxy-6-methylbenzoate (**38**) [48], methyl 4-hydroxy-2-methoxy-6-methylbenzoate (**39**) [53], methyl 4-hydroxybenzoate (**40**) [54], 7-acetyl-4,8-dihydroxy-6-methyl-1-tetralone (**41**) [8], benzoic acid (**42**) [55], 4-hydroxybenzaldehyde (**43**) [52], 4-hydroxybenzoic acid (**44**) [54], phenylacetic acid (**45**) [56]; 7 phenylpropanoids, 4-(3-hydroxypropyl)-2-methoxyphenol (**46**) [57], ferulic acid (**47**) [58], tetracosanyl ferulate (**48**) [59], 4-hydroxycinnamic acid (**49**) [52], methyl 4-hydroxycinnamate (**50**) [47], feruloyloxytetracosanoic acid (**51**) [60], phloiorubein (**52**) [61]; one lignan, (+)-pinoresinol (**53**) [62]; 7 steroids, β-siosterol (**54**) [63], stigmasterol (**55**) [63], β-sitostenone (**56**) [64], stigmasta-4,22-dien-3-one (**57**) [65], 6β-hydroxystigmast-4-en-3-one (**58**) [66], 3β-hydroxystigmast-5-en-7-one (**59**) [52], (22*E*)-3β-hydroxystigmasta-5,22-dien-7-one (**60**) [67]; one monoterpene, loliolide (**61**) [68]; one diterpene, phytol (**62**) [54]; four triterpenes, 3β,24ξ-dihydroxy-25-cycloartene (**63**) [69], 3β-hydroxy-11-oxo-12-ursene (**64**) [70], 9 (11),12-oleanadien-3-one (**65**) [71], 3β-hydroxy-20-oxo-30-norlupane (**66**) [70]; a carotenoid, β,ε-carotene-3,3′-diol (**67**) [72]; two alkaloids, indole-3-aldehyde (**68**) [73], indole-3-carboxylic acid (**69**) [74]; two fatty acids, linoleic acid (**70**) [75] and palmitic acid (**71**) [76] were readily identified by comparison of physical and spectroscopic data (UV, IR, ^1^H-NMR, [α]_D_, and mass spectrometry data) with values found in the literature.

### 2.2. Mushroom Tyrosinase Inhibitory Effect of Compounds Isolated from D. ensifolia

Tyrosinase is a copper-containing enzyme that catalyzes the oxidation of both monophenols (monophenolase activity) and diphenols (diphenolase activity) to the corresponding quinones and is responsible for the formation of melanin, which protects skin from the damage caused by UV radiation [20,24]. Excessive tyrosinase activity may lead to the overproduction or abnormal distribution of melanin, known as irregular hyperpigmentation of the skin. The use of tyrosinase inhibitors to prevent pigmentation is becoming increasingly important in the cosmetic and medicinal industries. These phenomena have motivated us to continue our research on natural tyrosinase inhibitors.

Compounds **1**–**4**, **8, 10**–**15, 20, 21,** and **41** were evaluated for their mushroom tyrosinase inhibitory effect. As shown in Table 3, compounds **11** and **14** exhibited potent concentration-dependent mushroom tyrosinase inhibitory activity, with IC_50_ values of 8.6 ± 0.8 and 14.5 ± 0.3 μM, respectively. However, the IC_50_ value of arbutin (positive control), a well-known whitening agent used in cosmetics, was 112.2 ± 5.4 μM. (2*S*)-2’,4’-Dihydroxy-7-methoxy-8-methylflavan (**11**) and (2*S*)-2’,4’-dihydroxy-7-methoxyflavan (**14**) showed approximately 13-fold and 7.7-fold stronger tyrosinase inhibitory activity than arbutin, respectively. Compounds **2**, **15**, **20**, and **21** also showed a more substantial inhibitory effect against mushroom tyrosinase than arbutin. Whereas, compounds **1**, **3**, **4**, **8**, **10**, **12**, **13**, and **41** had moderate or weak activity.

The results also noted interesting structure-activity relationships among the tyrosinase inhibitory activity of these tested compounds. The anti-tyrosinase activity of them can be ranked as follows: **11**, **14** >> **20**, **2** > **21** > **15** ≅ **41** ≅ arbutin > **8**, **1** ≅ **10** > **13**, **4** > **3**, **12**. Among flavonoids, it was found that compounds with a resorcinol skeleton in the ring B (e.g., **11** and **14**) showed the most potent anti-tyrosinase activity than that bearing a catechol moiety (e.g., **2** and **20**). As compounds with catechol moiety, the activity of **20** (a flavone) is better than **2** (a flavan), it was suggested that para-substituted hydroxy group in the B ring attached to an α,β-unsaturated carbonyl group of flavone forms a skeleton similar to that of tyrosine, thus, it showed better anti-tyrosinase activity than flavans. These relative activities are consistent with previous reports [77,78,79]. In addition, the position of resorcinol is also important for anti-tyrosinase activity. In comparison with **11**, compound **12** dramatically decreased the tyrosinase inhibitory effect; it can be conducted that compounds with 2′,4′-resorcinol moiety is essential for potent enzyme inhibitor than that of 3′,5′-resorcinol moiety. Introducing a non-hydroxy substituent at C-4′ (e.g., **3**, **4**, **12**, and **13**) notably reduced the tyrosinase inhibitory activity. Further experiments are needed to pinpoint the mechanism of this activity.

## 3. Materials and Methods

### 3.1. General

Silica gel *60 F_254_* precoated plates (*Merck*). Column chromatography (CC): silica gel (*Merk* 70–230 mesh). High-performance liquid chromatography (HPLC): *LDC Analytical-III* system; column: *Keystone spherisorb silica*, 5 μm, 250 × 10 mm. M.p.: *Yanaco**-MP-S3* micro-melting-point apparatus; uncorrected. Optical rotation: *Jasco-DIP-**1000* polarimeter; in methanol. UV Spectra: *Helios Beta* UV-Visible spectrophotometer; *λ*_max_ (log*ε*) in nm. IR Spectra: *Perkin-Elmer-**983G* FT-IR spectrophotometer; *ν* in cm^−1^. ^1^H-, ^13^C-, and 2D-NMR Spectra: *Varian-**Unity-Plus-400* and *Bruker-DMX-500* spectrometers; δ in ppm, *J* in Hz. EIMS and HREIMS: *Jeol-JMS-HX300* mass spectrometer; *m*/*z* (rel. %).

### 3.2. Plant Material

*Dianella ensifolia* were collected from Taipei, Taiwan, in July 2006 and positively identified by Y.-H. K. A voucher specimen (YHK 0671) has been deposited in the Herbarium of the College of Pharmacy, China Medical University, Taichung, Taiwan, R.O.C.

### 3.3. Extraction and Isolation

Dried roots (12.8 kg) of *Dianella ensifolia* were sliced and extracted with cold MeOH twice. After removal of the solvent under vacuum, the extract (376 g) was partitioned with EtOAc and water (1:1 *v*/*v*) three times and obtained EtOAc- and water-soluble fractions.

The EtOAc-soluble fraction (90 g) was chromatographed using silica gel (2.0 kg), using *n*-hexane, EtOAc, and MeOH of increasing polarity as eluent to give 10 fractions. Fraction 2 (7.6 g, *n*-hexane/EtOAc 95:5) was subjected to HPLC (*n*-hexane/EtOAc 9:1) to yield **3** (19.3 mg), **34** (583.2 mg), **38**, (23.8 mg), **48** (62.5 mg), and the mixture (25.2 mg) of **56** and **57**. Fraction 3 (10.1 g, *n*-hexane/EtOAc 90:10) was subjected to HPLC (CH_2_Cl_2_/EtOAc 97:3) to yield **7** (30.9 mg), **9** (85.5 mg), **24** (35.7 mg), the mixture (159.7 mg) of **54** and **55**, and **69** (5.32 g). Fraction 4 (4.6 g, *n*-hexane/EtOAc 80:20) was subjected to HPLC (CH_2_Cl_2_/EtOAc 85:15) to yield **1** (31.9 mg), **2** (26.7 mg), **4** (33.2 mg), **6** (26.6 mg), **10** (37.0 mg), **11** (28.1 mg), **12** (24.4 mg), **16** (25.5 mg), **23** (31.0 mg), **25** (25.2 mg), **28** (22.9 mg), **29** (38.2 mg), **39** (47.8 mg), **40** (46.7 mg), **58** (15.8 mg), and the mixture (18.3 mg) of **59** and **60**. Fraction 5 (4.6 g, *n*-hexane/EtOAc 70:30) was subjected to HPLC (CH_2_Cl_2_/EtOAc 85:15) to yield **5** (34.4 mg), **8** (33.2 mg), **26** (31.4 mg), **31** (14.1 mg), **35** (27.8 mg), **36** (18.4 mg), **41** (63.2 mg), **46** (16.5 mg), and **61** (15.6 mg). Fraction 6 (6.6 g, *n*-hexane/EtOAc 50:50) was subjected to HPLC (CH_2_Cl_2_/EtOAc 80:20) to yield **15** (51.8 mg), **20** (31.0 mg), **32** (28.9 mg), **33** (232.9 mg), **37** (22.8 mg), **47** (40.0 mg), **49** (21.6 mg), **53** (30.7 mg).

The dried aerial part of *D. ensifolia* (4.9 kg) was chopped and extracted with cold MeOH twice. The mixtures were filtered and concentrated to dryness under reduced pressure, producing a methanolic extract (331.0 g). The methanolic extract was partitioned with EtOAc and water (1:1, *v*/*v*) to obtain EtOAc- and water-soluble layers.

The EtOAc-soluble layer (185.0 g) was subjected to silica gel (5.0 kg) column chromatography with a gradient of *n*-hexane/ EtOAc /MeOH to obtain 11 fractions. Fraction 4 (35.3 g, *n*-hexane/EtOAc 90:10) was subjected to preparative HPLC, eluting with *n*-hexane/EtOAc (8:2) and *n*-hexane/CH_2_Cl_2_/EtOAc (5.5:4:0.5) to yield four steroids: **54** (1442 mg), **55** (961 mg), **56** (45.4 mg), **57** (31.5 mg). Fraction 5 (9.6 g, *n*-hexane/EtOAc 80:20) was subjected to preparative HPLC, eluting with *n*-hexane/EtOAc (6:4), CH_2_Cl_2_/EtOAc (8.5/1.5), and *n*-hexane/CH_2_Cl_2_/EtOAc (5.5:4:0.5) to yield three flavonoids: **8** (10.8 mg), **9** (21.6 mg), and **18** (6.7 mg); six aromatics: **38** (1.3 mg), **39** (2.4 mg), **40** (22.2 mg), **43** (2.2 mg), **50** (2.9 mg), and **52** (6.0 mg); four triterpenoids: **63** (7.0 mg), **64** (6.6 mg), **65** (4.8 mg), and **66** (11.5 mg); one chromone: **29** (7.7 mg), one diterpene: **62** (21.2 mg) and a fatty acid: **71** (18.5 mg). Fraction 6 (8.9 g, *n*-hexane/EtOAc 70:30) was purified by HPLC, eluting with *n*-hexane/EtOAc (7:3) to obtain **17** (4.8 mg), follow by CH_2_Cl_2_/EtOAc (9.5:0.5 to 8:2) to yield **1** (3.1 mg), **10** (5.2 mg), **11** (12.1 mg), **12** (27.5 mg), **13** (29.7 mg), **14** (33.3 mg), and **16** (54.2 mg) was purified by *n*-hexane/EtOAc (6:4). Total are eight flavonoids isolated from fraction 6. In addition, four aromatics, **36** (11.8 mg), **41** (9.4 mg), **42** (5.3 mg), **51** (2.5 mg), were obtained by eluting with *n*-hexane/EtOAc (6.5:3.5 to 6:4) and CH_2_Cl_2_/EtOAc (8.5:1.5 to 8:2). One chromone, **44** (25.3 mg), was obtained by eluting with *n*-hexane/CH_2_Cl_2_/EtOAc (5.5:4:0.5). One chalcone, **27** (3.1 mg) was obtained by eluting with *n*-hexane/EtOAc (6:4). A carotenoid, ***67*** (1.7 mg), was obtained by eluting with *n*-hexane/EtOAc (7:3) and CH_2_Cl_2_/EtOAc (9.5:0.5 to 8:2). Fraction 7 (12.4 g, *n*-hexane/EtOAc 50:50) was purified by HPLC, eluting with *n*-hexane/EtOAc (6:4 to 1:1) and CH_2_Cl_2_/EtOAc (9.5:0.5 to 1:1) to yield four flavonoids [**15** (10.1 mg), **21** (9.4 mg), **22** (4.7 mg), **19** (29.6 mg)], seven aromatics [**29** (7.7 mg), **30** (25.3 mg), **49** (63.8 mg), **37** (13.0 mg), one chalcone [**26** (23.1 mg)], two alkaloids [**68** (9.0 mg), **69** (11.6 mg)], one lignan [**53** (17.8 mg)], and one monoterpene [**61** (79.8 mg)].

### 3.4. (2S)-4’-Hydroxy-6,7-Dimethoxyflavan (***1***)

Yellow oil; ${\left[\alpha \right]}_{D}^{25}$ −17.1° (*c* 0.01, MeOH); UV (MeOH) *λ*_max_ (log *ε*): 286 (4.12), 225 (4.66) nm; IR (neat) *ν*_max_: 3424, 3010, 2935, 2859, 1605, 1512, 1214, 1116 cm^−1^; EIMS: *m/z* (rel. int.): 286 (M^+^, 100), 180 (8), 167 (41), 166 (13), 120 (7); HREIMS: 286.1209 (C_17_H_18_O_4_^+^, calc. 286.1205); ^1^H-NMR: see Table 1; ^13^C-NMR: see Table 2.

### 3.5. (2S)-3’,4’-Dihydroxy-7-Methoxy-8-Methylflavan (***2***)

Brown oil; ${\left[\alpha \right]}_{D}^{25}$ −15.8° (*c* 0.01, MeOH); UV (MeOH) *λ*_max_ (log *ε*): 282 (3.66), 227 (4.08) nm; IR (neat) *ν*_max_: 3403, 3090, 2952, 2853, 1619, 1510, 1454, 1275, 1162 cm^−1^; EIMS: *m/z* (rel. int.): 286 (M^+^, 77), 269 (13), 164 (29), 151 (100), 136 (34); HREIMS: 286.1207 (C_17_H_18_O_4_^+^, calc. 286.1205); ^1^H-NMR: see Table 1; ^13^C-NMR: see Table 2.

### 3.6. (2S)-2’-Hydroxy-7-Methoxyflavan (***3***)

Brown oil; ${\left[\alpha \right]}_{D}^{25}$ −26.0° (*c* 0.01, MeOH); UV (MeOH) *λ*_max_ (log *ε*): 280 (3.84), 221 (4.25) nm; IR (neat) *ν*_max_: 3368, 3012, 2927, 2853, 1607, 1513 cm^−1^; EIMS: *m/z* (rel. int.): 256 (M^+^, 34), 239 (50), 150 (28), 137 (58), 73 (26), 59 (100); HREIMS: 256.1104 (C_16_H_16_O_3_^+^, calc. 256.1099); ^1^H-NMR: see Table 1; ^13^C-NMR: see Table 2.

### 3.7. 4-Hydroxy-4-(7-Methoxy-8-Methylchroman-2-yl)-Cyclohex-2-Enone (***4***)

Colorless oil; ${\left[\alpha \right]}_{D}^{25}$ +30.8° (*c* 0.01, MeOH); UV (MeOH) *λ*_max_ (log *ε*): 281 (3.51), 227 (4.48) nm; IR (neat) *ν*_max_: 3435, 3044, 2935, 1676, 1601, 1493, 1434, 1113 cm^−1^; EIMS: *m/z* (rel. int.): 288 (M^+^, 29), 270 (5), 178 (13), 177 (100), 112 (33); HREIMS: 288.1363 (C_17_H_20_O_4_^+^, calc. 288.1362); CD (MeOH): Δ*ε*_392_ 0.47, Δ*ε*_334_ 1.10 (dm^3^ mol^−1^ cm^−1^); ^1^H-NMR: see Table 1; ^13^C-NMR: see Table 2.

### 3.8. In Vitro Mushroom Tyrosinase Inhibition Assay

The mushroom tyrosinase inhibitory assay has been described in detail previously [79]. Briefly, various concentrations of test compounds (100 μL) and 2 mM l-tyrosine (80 μL, Sigma-Aldrich, St. Louis, MO, USA) were mixed in 96-well plates. 20 μL of mushroom tyrosinase (1000 U/mL, Sigma-Aldrich, USA) was added to initiate the assay. The absorbance of the mixture was measured at 490 nm using a microplate reader (μQuant™, BioTek Winooski, VT, USA). The percentage inhibition of mushroom tyrosinase was calculated according to the following equation: Inhibition (%) = [1 − (S − SB)/(C − CB)] × 100%, where S, SB, C, and CB are the absorbance of the sample, the blank sample, the control, and the blank control, respectively.

## 4. Conclusions

Four new flavans, (2*S*)-4’-hydroxy-6,7-dimethoxyflavan (**1**), (2*S*)-3’,4’-dihydroxy-7-methoxy-8-methylflavan (**2**), (2*S*)-2’-hydroxy-7-methoxyflavan (**3**), and (2*S*,1’*S*)-4-hydroxy-4-(7-methoxy-8-methylchroman-2-yl)-cyclohex-2-enone (**4**), along with 67 known compounds were isolated from whole plants (roots and the aerial part) of *D. ensifolia*. Selected compounds of *D. ensifolia* display anti-tyrosinase properties. The presence of 2′,4′-resorcinol (**11** and **14**) or catechol (**2** and **20**) moiety in the ring B of flavonoids appears to be required for the anti-tyrosinase activity. Although the detailed mechanism of action of these compounds remains to be determined, the results confirmed that *D. ensifolia* is a valuable source of herbal medicine from which natural-based whitening agents can be derived.

## Figures and Tables

**Figure 1 plants-11-02142-f001:**
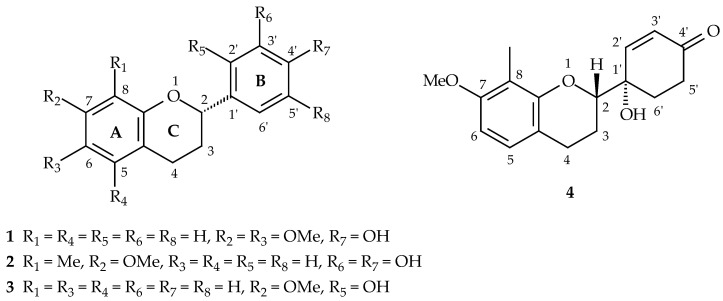
The chemical structures of new flavans (**1**–**4**).

**Figure 2 plants-11-02142-f002:**
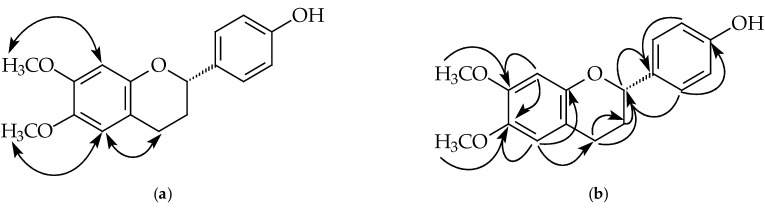
Key NOESY contacts (**a**) and HMBC connectivities (**b**) of compound **1**.

**Figure 3 plants-11-02142-f003:**
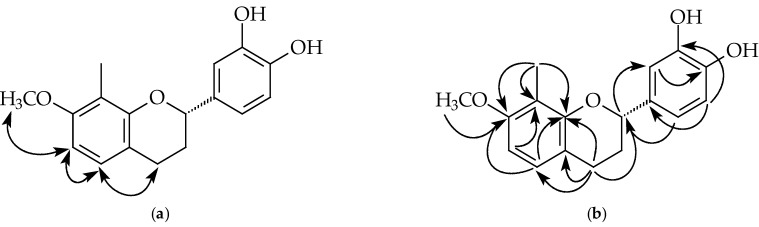
Key NOESY contacts (**a**) and HMBC connectivities (**b**) of compound **2**.

**Figure 4 plants-11-02142-f004:**
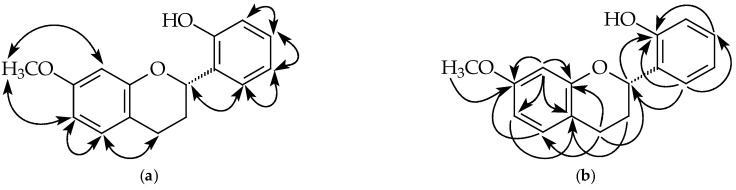
Key NOESY contacts (**a**) and HMBC connectivities (**b**) of compound **3**.

**Figure 5 plants-11-02142-f005:**
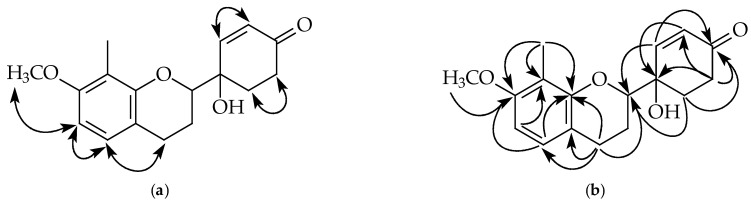
Key NOESY contacts (**a**) and HMBC connectivities (**b**) of compound **4**.

**Figure 6 plants-11-02142-f006:**
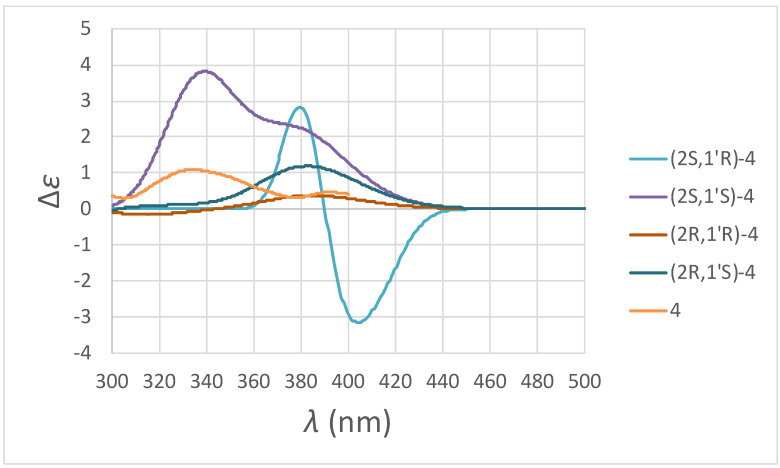
Experimental and calculated ECD spectra of compound **4**.

**Table 1 plants-11-02142-t001:** ^1^H-NMR data (CDCl_3_) of compounds **1**–**4**. Chemical shifts δ in ppm, *J* in Hz.

H	1 ^a^	2 ^b^	3 ^b^	4 ^a^
2	4.92 (dd, *J* = 10.2, 2.6)	4.96 (dd, *J* = 10.5, 2.4)	5.18 (dd, *J* = 10.2, 2.7)	4.01 (dd, *J* = 11.2, 2.0)
3	2.05 (m) 2.14 (m)	1.90 (m) 2.16 (m)	2.26 (m)	1.92 (m) 2.04 (m)
4	2.70 (ddd, *J* = 16.4, 5.2, 2.8, H_β_) 2.92 (ddd, *J* = 16.4, 10.8, 5.4, H_α_)	2.72 (ddd, *J* = 15.2, 4.8, 4.0, H_β_) 2.92 (m, H_α_)	2.80 (ddd, *J* = 16.3, 5.0, 2.5, H_β_) 2.96 (m, H_α_)	2.81 (m)
5	6.57 (s)	6.85 (d, *J* = 8.0)	7.00 (d, *J* = 8.4)	6.86 (d, *J* = 8.2)
6	–	6.44 (d, *J* = 8.0)	6.52 (dd, *J* = 8.4, 2.4)	6.46 (d, *J* = 8.2)
7	–	–	–	–
8	6.46 (s)	–	6.46 (d, *J* = 2.4)	–
2’	7.28 (d, *J* = 8.4)	6.95 (s)	–	7.09 (dd, *J* = 10.4, 1.2)
3’	6.83 (d, *J* = 8.4)	–	6.91 (m)	6.06 (d, *J* = 10.4)
4’	–	–	7.20 (m)	–
5’	6.83 (d, *J* = 8.4)	6.84 (s)	6.90 (m)	2.49 (dd, *J* = 17.2, 5.6) 2.81 (m)
6’	7.28 (d, *J* = 8.4)	6.84 (s)	7.14 (dd, *J* = 8.3, 1.2)	2.16 (m)
MeO-6	3.82 (s)	–	–	–
MeO-7	3.80 (s)	3.80 (s)	3.75 (s)	3.79 (s)
Me-8	–	2.20 (s)	–	2.08 (s)
OH	5.05 (br s)	5.23 (br s), 5.25 (br s)		

^a^ Measured at 400 MHz. ^b^ Measured at 500 MHz.

**Table 2 plants-11-02142-t002:** ^1^^3^C-NMR data (CDCl_3_) of compounds **1**–**4**. Chemical shifts δ in ppm.

C	1 ^a^	2 ^b^	3 ^b^	4 ^a^
2	77.5	77.2	78.5	80.5
3	30.3	30.4	28.1	21.7
4	25.2	25.1	24.4	24.7
4a	105.3	113.4	114.1	113.3
5	111.9	125.8	130.2	125.9
6	142.5	102.8	108.5	103.4
7	148.5	156.1	159.1	156.1
8	100.8	114.0	101.8	113.8
8a	147.7	152.8	154.4	151.8
1’	133.5	134.9	125.5	70.5
2’	127.3	112.8	154.7	148.7
3’	115.0	142.9	117.2	129.5
4’	154.7	142.4	129.4	198.2
5’	115.0	114.9	120.4	33.9
6’	127.3	118.2	127.0	30.9
MeO-6	56.6	–	–	–
MeO-7	56.0	55.9	55.4	55.9
Me-8	–	8.8	–	9.0

^a^ Measured at 100 MHz. ^b^ Measured at 125 MHz.

**Table 3 plants-11-02142-t003:** Mushroom tyrosinase inhibitory effect of tested compounds.

Compound	IC_50_ (μM) ^a^	Compound	IC_50_ (μM) ^a^
**1**	170.9 ± 4.6	**13**	261.2 ± 4.5
**2**	68.9 ± 5.7	**14**	14.5 ± 0.3
**3**	>300	**15**	108.6 ± 6.2
**4**	275.9 ± 8.4	**20**	41.8 ± 3.3
**8**	147.5 ± 6.5	**21**	92.3 ± 4.2
**10**	175.9 ± 10.7	**41**	111.6 ± 7.1
**11**	8.6 ± 0.8	Arbutin ^b^	112.2 ± 5.4
**12**	>300		

^a^ IC_50_ is the concentration of the sample required to inhibit 50% of the enzyme. ^b^ Positive control.

## Data Availability

The data presented in this study are available in the supplementary material.

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
