# Peer review of "Tyrosinase Inhibitors Derived from Chemical Constituents of *Dianella ensifolia"

_plants, 2022, doi:10.3390/plants11162142_

Round 1
Reviewer 1 Report
Dear authors,
The article entitled: "Tyrosinase inhibitors derived from chemical constituents of 2 Dianella ensifolia" is of great interest and investigates the potential inhibitory effect of newly isolated compounds. After a round of minor revisions, the article can be considered for publication.
Line 28: has the potential to be used as a herbal medicine due to its chemical constituents
Line 63: The role of melanin is to protect the skin against ultraviolet damage
Author Response
Dear Reviewer,
Thank you very much for your precious time in reviewing our paper and providing comments. The sentence has been revised as suggested by your comments. [Pg1, Line29; Pg2, Line63].
After carefully checking the contents and all references, some corrections have been marked up using the "Track Changes" function in the revised version.
Best regards,
Horng-Huey Ko

Reviewer 2 Report
The paper of Chen et al. is a very good paper regarding nature chemical compounds with beneficial effects for human health. The methods of extraction and characterization of the new 4 flavans isolated from Daniella ensifolia were very well chosen and described. This paper can be published in Plant journal in this form.
Author Response
Dear Reviewer,
Thank you for your precious time and positive feedback on our article.
Sincerely yours,
Horng-Huey Ko
Reviewer 3 Report
Attach file

Author Response
Dear Reviewer,
Thank you very much for your precious time in reviewing our paper and providing comments. Our manuscript has been revised as suggested by your comments. Below we provide the point-by-point responses. After carefully checking the contents and all references, some corrections have been marked up using the "Track Changes" function in the revised version.
- p.1, stereochemistry Sline 30-32: (2S)-4'-hydroxy-6,7-dimethoxyflavan.... changed by (2S)-4'-hydroxy-6,7-dimethoxyflavan and other compounds in the abstract.
Response: Revised accordingly.
2. p.2, line 89 and 97:....(EtOAc-)....changed by ......(EtOAc)...
Response: Revised accordingly.
3. p.3, 4, 5, 6: In the Figures 2, 3, 4, 5 and 6 the connectivities arrows in NOESY and HMBC of structures are not configured.
Response: The connectivities arrows in NOESY and HMBC of structures have been modified.
4. p10, lines 318, 323, 345 and 346: Change S configuration by italic: S configuration.
Response: Revised accordingly.
Sincerely yours,
Horng-Huey Ko
